# Income inequality and its relationship with loneliness prevalence: A cross-sectional study among older adults in the US and 16 European countries

**Thamara Tapia-Muñoz**[1,2,3,4,5], **Ursula M. Staudinger**[6,7], **Kasim Allel**[5,8,9], **Andrew Steptoe**[1], **Claudia Miranda-Castillo**[4,10,11], **José T. Medina**[2,3,5], **Esteban Calvo**[2,3,5,6]*

1 Department of Behavioural Science and Health, University College London, London, United Kingdom, 2 Society and Health Research Center, School of Public Health, Universidad Mayor, Santiago, Chile, 3 Laboratory on Aging and Social Epidemiology, Facultad de Ciencias Sociales y Artes, Universidad Mayor, Santiago, Chile, 4 Millennium Institute for Caregiving Research, Santiago, Chile, 5 Millennium Nucleus on Sociomedicine, Santiago, Chile, 6 Mailman School of Public Health, Columbia University, New York, NY, United States of America, 7 Technische Universität Dresden, Dresden, Germany, 8 Institute for Global Health, University College London, London, United Kingdom, 9 College of Medicine and Health, University of Exeter, Exeter, United Kingdom, 10 Faculty of Nursing, Universidad Andres Bello, Santiago, Chile, 11 Millennium Institute for Research in Depression and Personality, Santiago, Chile

* esteban.calvo@columbia.edu

## Abstract

### Backgrounds

The prevalence of loneliness increases among older adults, varies across countries, and is related to within-country socioeconomic, psychosocial, and health factors. The 2000–2019 pooled prevalence of loneliness among adults 60 years and older went from 5.2% in Northern Europe to 24% in Eastern Europe, while in the US was 56% in 2012. The relationship between country-level factors and loneliness, however, has been underexplored. Because income inequality shapes material conditions and relative social deprivation and has been related to loneliness in 11 European countries, we expected a relationship between income inequality and loneliness in the US and 16 European countries.

### Methods

We used secondary cross-sectional data for 75,891 adults age 50+ from HRS (US 2014), ELSA (England, 2014), and SHARE (15 European countries, 2013). Loneliness was measured using the R-UCLA three-item scale. We employed hierarchical logistic regressions to analyse whether income inequality (GINI coefficient) was associated with loneliness prevalence.

### Results

The prevalence of loneliness was 25.32% in the US (HRS), 17.55% in England (ELSA) and ranged from 5.12% to 20.15% in European countries (SHARE). Older adults living in countries with higher income inequality were more likely to report loneliness, even after adjusting

**Data Availability Statement:** The data underlying the results presented in the study are available

from: https://hrs.isr.umich.edu/ http://www.share-project.org/ https://www.elsa-project.ac.uk/.

**Funding:** (EC) Research and Development National Agency– FONDECYT REGULAR-1140107 and Research and Development National Agency– Millennium Science Initiative Program - Millennium Nucleus on Sociomedicine - NCS2021_013 https://www.anid.cl/ https://www.iniciativamilenio.cl/en/home_en/ (TT) Scholarship from the Chilean Ministry of Education and Research and Development National Agency https://www.anid.cl/ (CM) Research and Development National Agency - Millennium Science Initiative Program – ICS2019_024 and ICS13_005 and ANID-FONDECYT-1191726. https://www.anid.cl/ https://www.iniciativamilenio.cl/en/home_en/ The funders had no role in study design, data collection and analysis, decision to publish, or preparation of the manuscript.

**Competing interests:** The authors have declared that no competing interests exist.

for the sociodemographic composition of the countries and their Gross Domestic Products per capita (OR: 1.52; 95% CI: 1.17–1.97).

## Discussion

Greater country-level income inequality was associated with higher prevalence of loneliness over and above individual-level sociodemographics. The present study is the first attempt to explore income inequality as a predictor of loneliness prevalence among older adults in the US and 16 European countries. Addressing income distribution and the underlying experience of relative deprivation might be an opportunity to improve older adults' life expectancy and wellbeing by reducing loneliness prevalence.

## 1. Introduction

Scholars and policymakers worldwide have expressed growing concerns about loneliness, especially among young and older adults [1, 2]. Loneliness has been defined as a negative emotional experience produced by the discrepancy between the desired social and emotional life and the one taking place [3]. People can be socially connected and still feel lonely [4]. When loneliness is intense and frequently experienced (chronic loneliness) [5], it can have severe health consequences for older adults [4, 6–18]. Firstly, loneliness is associated with increased all-cause mortality, and it is a risk factor for suicide [4, 9, 13, 15–17, 19]. Secondly, loneliness impacts older adults' mental health, with lonelier people experiencing higher rates of depression and anxiety and a poorer quality of life [20–25]. Thirdly, it is a risk factor for dementia and other causes of disability [26, 27].

There are cross-country differences in loneliness prevalence. The diversity of loneliness measures and varying cut-off points for the same scales have led to differences in loneliness prevalence [6, 28, 29]. However, holding measures and cut-off points constant, there are still sizable cross-country variations in the prevalence of loneliness [7, 28, 30, 31]. Yang and Victor (2011) studied loneliness prevalence in 24 European countries. Loneliness was divided into "sometimes lonely" and "frequent loneliness" (those who reported loneliness "all the time" or "most of the time"). Countries were divided into three groups based on the author's assessment of the relationship for each country. The first group encompassed Bulgaria, Hungary, Latvia, Poland, Romania, Russia, Slovakia, and Ukraine, with loneliness ratings ranging from 18.8% in Romania to 34% in Ukraine, which had the highest prevalence of loneliness. Group two, composed of Belgium, Denmark, Finland, Germany, Ireland, Netherlands, Norway, Sweden, Switzerland, and the United Kingdom, had the lowest "frequent loneliness" prevalence among adults 60 years and older, with percentages below 10%. Finally, the third group, composed of Austria, Cyprus, Estonia, France, Portugal, Slovenia, and Spain, ranged from 10% in Cyprus to 15% in Slovenia for adults over 60 years old [28]. In a pooled analysis of studies conducted between 2000 and 2019 measuring loneliness in people 60 years and older, the lowest prevalence of loneliness was 5.2% in Northern Europe (Finland, Norway, Sweden, Denmark), and the highest prevalence was 24.2% in Eastern Europe (Belarus, Estonia, Hungary, Latvia, Moldova, Poland, Romania, Russia, Slovenia, Ukraine) [31]. Moreover, in the US Health and Retirement Study (HRS) (wave 2012), loneliness among people 60 years and older was 56.63% when using the responses "some of the time" or "often" to any of the three statements in the revised version of The University of California Los Angeles Loneliness Scale (R-UCLA scale) and 37.08% when using the responses "some of the time" or "often" to at least two out of the three items [6].

The theoretical models addressing cross-country differences in loneliness have pointed out that loneliness is a complex phenomenon with genetic, social, and environmental contributors [32–34]. Individual-level factors related to loneliness have been more frequently considered in interventions addressing loneliness [2]; however, these interventions are not entirely effective [35]. Targeting structural elements might be needed to overcome the unequal social conditions of older adults with, among other consequences, a reduction in individual loneliness [36]. Differences in population-level sociodemographic composition across countries seems to play an important role in explaining cross-country differences in loneliness. Marital status, age, educational level and health status are individual-level factors that may contribute to cross-country differences in loneliness [1, 37], but little is known about the relationship between country-level aggregate factors and loneliness. A few published studies have focused on cultural factors to explain cross-country variations in loneliness, noting factors such as presence of multigenerational households and connections [30, 38]. The Fokkema, De Jong Gierveld & Dykstra [34] model has highlighted the importance of interactions between individual and societal factors like older adults' living conditions. Lately, a relationship between neighbourhood social deprivation and loneliness has been observed in the UK, in which more socially deprived areas reported higher levels of individual and area-based loneliness [39].

The plan for the Decade of Healthy Ageing (2020–2030) established an action item to identify and tackle loneliness through a community-based approach that offers older adults equal opportunities for leisure and social activities [40, 41]. The plan is linked to the Sustainable Developed Goals for the decade, which call for a united front to overcome inequality and ensure healthy ageing for all older people regardless of residency, gender, ethnicity, level of education, civil status, and health condition [42]. The Marmot reports on health inequalities have shown that income inequality within a country produces differences in material conditions and increases relative deprivation. Relative deprivation is the psychological effect of income inequality on people [43, 44]. The social gradients determine access to education, jobs, proper incomes, wealth, and, at the same time, increase insecurity, anxiety, social isolation, among other mental health outcomes. More unequal countries have a higher social gradient; therefore, factors which represent social position such as gender, race and ethnicity, education, and occupation, are more impactful [43–45]. Recently, using data from eleven countries that were part of the fifth and sixth waves of the Survey of Health, Ageing and Retirement in Europe (SHARE) (Austria, Belgium, Czech Republic, Denmark, Estonia, France, Germany, Italy, Slovenia, Spain and Sweden), a study reported a relationship between country inequality (GINI Index) and loneliness [46]. Further evidence of the relationship between income inequality and the prevalence of loneliness among the older population can provide information about the extent of country-level factors' contributions to cross-country differences in loneliness and their potential roles in the success of loneliness interventions [1, 47]. Hence, the current study aimed to explore the relationship between country-level income inequality and the prevalence of loneliness in the USA and 16 European countries.

## 2. Materials and methods

### 2.1 Study design

This is a cross-sectional observational study of secondary 2013 and 2014 data from nationally representative surveys of older adults.

### 2.2 Study population and analytic sample

We drew data for 75,891 older adults aged 50 and older from the United States and 16 European countries from three well-characterized cohorts: the Health and Retirement Study (HRS;

United States) wave 12 collected in 2014 [48], the English Longitudinal Study of Aging (ELSA; England) wave seven collected between 2014 and 2015 [49], and The Survey of Health, Ageing and Retirement in Europe (SHARE: Austria, Belgium, Czech Republic, Denmark, Estonia, France, Germany, Israel, Italy, Luxemburg, Netherlands, Slovenia, Spain, Sweden, and Switzerland) corresponding to wave five measured in 2013 [50]. The data harmonization process has been described in detail elsewhere [51–53].

The eligibility criteria were defined by the following. First, from the total populations represented by the surveys, participants were considered eligible when they met the criteria to enter the wave, were reported alive and responded to the survey. Second, we dropped participants who were partners of main respondents and younger than 50 years. The three survey's methodological protocols consider complete cases when there is information for two out of the three questions of the R-UCLA. Accordingly, we dropped participants from the study when the three R-UCLA items were missing (see Fig 1). The study's analytic sample was built using information from the three-item R-UCLA loneliness scale and the complete cases for all the covariates (S1 Table). The missing values among the independent variables were around 1%, therefore the data was not imputed (S2 Table). Moreover, we employed robustness checks to avoid potential biases (S3 Table).

ELSA waves were reviewed and approved by the National Research Ethics Service (London Multicentre Research Ethics Committee). From the wave 4, SHARE was reviewed and approved by the Ethics Council of the Max Planck Society. Finally, the University of Michigan Institutional Review Board reviewed and approved HRS waves. All participants gave informed consent.

## 2.3 Variables

Our main outcome was loneliness prevalence, measured using three out of the 20 items detailed in the R-UCLA loneliness scale. The items asked how often the person feels "left out,"

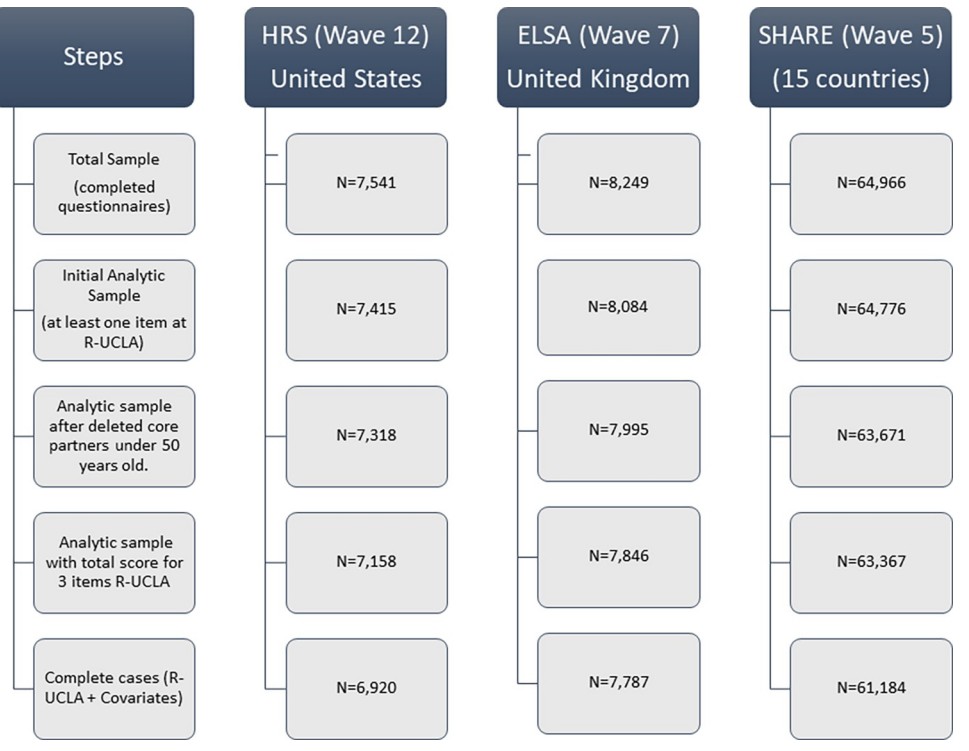

**Fig 1. Participants flow chart.**

"isolated from others," and "lacking companionship" [54]. For each item, the scale of the responses was 1 = hardly ever or never, 2 = sometimes and 3 = often. For each participant, we calculated a sum score for the *intensity* of loneliness, ranging from 3 to 9. Among older adults, the original intensity scale had a unidimensional structure, high reliability, an alpha Cronbach of 0.89, and a test-retest coefficient of 0.73 [54]. For the present study, the average alpha Cronbach was 0.77 (HRS = 0.81, ELSA = 0.83, and SHARE = 0.75; S4 Table).

To measure the *prevalence* of loneliness we followed the 6-point cut-off previously established by Steptoe et al. [16].

### 2.4 Country variables

Our primary exposure was country-level income inequality, measured with the 2013 or 2014 GINI index reported by the World Bank [55]. The theoretical values of the GINI range between 0 and 100, with higher numbers indicating higher inequality.

Considering the relationship between economic growth and income inequality, we used the 2013 or 2014 gross domestic product per capita adjusted by power purchase parity (GDP-PPP) as an independent variable [56]. We used a random country effect to adjust our estimates for unobserved country characteristics (the United States was the reference country). To facilitate interpretation, GDP and GINI were standardized.

### 2.5 Individual-level variables

Based on the model defined by Fokkema, De Jong Gierveld & Dykstra [33], we considered the following individual level factors for the prevalence of loneliness: participant age in years at baseline as a continuous variable and gender as participant self-classification of sex, coded as woman or man. Due to the small number of participants above 90 years (n = 600), all people over the age of 90 were recoded as 90 years. We also adjusted the statistical models for marital status (legally married or *de facto* partnered, separated or divorced, widowed, or single never married) and level of education (higher education versus no higher education). Work status measured paid work (full- or part-time, salaried, or self-employed, combined or with partial retirement) as opposed to not working for pay (complete retirement, disabled, unemployed, or out of the labour force). Moreover, we considered health factors. Self-reported health was obtained from the self-rate item about the state of health. The responses range from 1 = Poor to 5 = Excellent. Functional limitations were assessed using the three items (bathe, dress, and eat) defined in the Wallace and Herzog measure Activities of daily living (ADLs). Pain was obtained from the dichotomic item for being troubled with pain often (yes/no) and depressive mood comes from the statement "I have felt depressed" of the CED-S and EURO-D questionnaires and was used as a proxy for depression given the differences between surveys.

### 2.6 Statistical analysis

First, we performed descriptive analyses to characterise participants and countries. Next, we ran hierarchical logistic models to estimate the relationship between country economic inequality and the prevalence of loneliness within individuals nested within countries. Using logistic regression models, we examined the unadjusted relationship between the individual-level covariates and the prevalence of loneliness (Table B in S1 File). A random slope in age was used in the hierarchical models due to cross-country variations in the unadjusted relationship between age and loneliness (Fig A in S1 File).

We computed four sequential models to analyse the relationship between country-level economic inequality and individual-level loneliness. Model 1 included a fixed and random intercept only, allowing for an estimation of Intra-Class Correlation (ICC). Model 2 included the

**Table 1. Participants sociodemographic and health characteristics (N = 75,891).**

| | Mean (SD) |
|---|---|
| Age (years) | 66.63 (9.76) |
| Self-Reported Health (total score) | 2.93 (1.08) |
| | **Frequency (%)** |
| Gender | |
| Men | 33,632 (44.32) |
| Women | 42,259 (55.68) |
| Marital Status | |
| Married or partnered | 56,261 (74.13) |
| Divorced or separated | 6,366 (8.39) |
| Widowed | 9,914 (13.06) |
| Single never married | 3,350 (4.41) |
| Educational Attainment | |
| Less than college | 58,689 (77,43) |
| College and above | 17,202 (22.67) |
| Work status | |
| No Worker | 53,916 (71.04) |
| Worker | 21,975 (28.96) |
| Functional limitations | |
| No limitation | 68,280 (89,97) |
| Low limitation | 4,711 (6.21) |
| Moderate limitation | 2,123 (2.80) |
| Severe limitation | 777 (1.02) |
| Depressive Mood | |
| No | 50,465 (66.50) |
| Yes | 25,426 (33.50) |
| Pain Presence | |
| No | 44,246 (58,30) |
| Yes | 31,645 (41.70) |

GINI index, allowing for an unadjusted estimation of its relationship with loneliness. Model 3 added individual-level control variables to model 2. Finally, model 4 added GDP per capita as a country-level control variable to model 3. Considering the total variance explained by the Akaike information criterion (AIC) and the Bayesian information criterion (BIC), Model 3 was the best solution for loneliness prevalence (see results section, Table 2). Considering that less than 30 clusters might affect the estimation of random effect errors [57], we also performed a regression model using bootstrap error with 100 iterations. The model yielded the same results, which can be found in the supplementary materials, section 4, S3 Table.

The final equation to predict the prevalence of loneliness is formalised in the following equation:

$$
\begin{aligned}
\text{Log} (Y_{ij}) = \beta_{00} + \beta_1 {*} \text{Age}_{ij} + \beta_2 {*} \text{Gender}_{ij} + \beta_3 {*} \text{Marital Status}_{ij} + \beta_4 {*} \text{Educational} \\
\text{Attainment}_{ij} + \beta_5 {*} \text{Work status}_{ij} + \beta_6 {*} \text{Functional limitations}_{ij} + \beta_7 {*} \text{Depressive} \\
\text{Mood}_{ij} + \beta_8 {*} \text{Self} - \text{reported Health}_{ij} + \beta_9 {*} \text{Pain prevalence}_{ij} + \beta_{10} {*} \text{GINI}_j + b_{0j}^{*} + \\
b_{1j}^{*} {*} \text{Grand} - \text{mean centered age}_{ij} + \varepsilon_{ij}
\end{aligned} \quad \text{Eq(1)}
$$

Where $\log(Y_{i,j})$ is the expected prevalence of loneliness; $\beta_{00}$ is the odds of loneliness in an

**Table 2. Country-level descriptive statistics (n = 17).**

| Survey | Country (N) | Loneliness | GINI | GDP (US $) |
|---|---|---|---|---|
| | | n (%) | Mean (Sd) | Mean (Sd) |
| | **Group** | **10,004 (13.18)** | **32.37(4.32)** | **42,608.92 (11529.04) IQR = 15,318** |
| | | | IQR = 6.6 | |
| HRS | US | 1,752 (25.32) | 41 | 55,033 |
| ELSA | England | 1,367 (17.55) | 33.2 | 40,868 |
| SHARE | Spain | 573 (9.63) | 36.2 | 32,604 |
| | Germany | 480 (8.90) | 31.1 | 45,232 |
| | Estonia | 842 (15.64) | 35.1 | 27,496 |
| | Belgium | 707 (13.45) | 27.7 | 43,611 |
| | Czech Republic | 925 (17.81) | 26.5 | 30,486 |
| | Italy | 931 (20.15) | 34.9 | 36,131 |
| | Sweden | 263 (6.08) | 28.8 | 45,722 |
| | France | 479 (11.34) | 32.5 | 39,524 |
| | Austria | 260 (6.53) | 30.8 | 47,922 |
| | Netherlands | 289 (7.46) | 28.1 | 49,242 |
| | Denmark | 199 (5.12) | 28.5 | 46,727 |
| | Switzerland | 1875 (6.06) | 32.5 | 60,109 |
| | Slovenia | 248 (8.84) | 26.2 | 29,797 |
| | Israel | 380 (18.00) | 39.8 | 34,179 |
| | Luxemburg | 178 (11.81) | 32 | 95,591 |

average country; $x_{ij}$ are individual-level predictors of loneliness in the country j; $w_j$ is the country-level variable (GINI Index), $b_{0j}^*$ is the country-specific deviations around the OR for the prevalence of loneliness; $b_{1j}^*$ is the random slope in age; and $\varepsilon_{i,j}$ is the error term of the observed Logit ($Y_{i,j}$). All analyses were performed in Stata version 16.0 [58], using the command "xtmelogit" and a 95% confidence level.

## 3. Results

### 3.1 Descriptive results

Out of all participants, 56% were female and the mean age was 67 years (SD = 9.76). Other characteristics are described in Table 1.

Table 2 shows descriptive statistics for loneliness, GINI (group mean = 32.37; SD = 4.32; IQR = 6.6) and GDP-PPP per capita (group mean = 42,609; SD = 11,529; IQR = 15,318). There was substantial variation in the prevalence of loneliness between countries. The prevalence was 25.64% in the US (HRS), 17.60% in England (ELSA) and 5.22% to 20.15% in SHARE countries.

### 3.2 Hierarchical regression models (HLM) results in the prevalence of loneliness

HLM results are reported in Table 3. The unadjusted relationship between individual-level variables and loneliness prevalence was statistically significant (Table B in S1 File). As indicated by the Intra-Class Correlation (ICC), the variability between countries accounted for 7.9% of the total variation in the likelihood of an individual being lonely. In an average country, the odds of being lonely, defined as scoring more than 6 points in the three items of R-UCLA, was 0.13. However, there was statistically significant variability in the odds of loneliness between countries (Between country variance = 0.283; 95% IC: 0.144–0.559).

**Table 3. Hierarchical logistic model for the prevalence of loneliness (n = 75,891).**

| | Model 1 | | | Model 2 | | | Model 3 | | | Model 4 | | |
|---|---|---|---|---|---|---|---|---|---|---|---|---|
| | OR | 95% CI | | OR | 95% CI | | OR | 95% CI | | OR | 95% CI | |
| **Fixed Effects** | | | | | | | | | | | | |
| Constant | 0.13** | 0.1 | 0.16 | 0.10** | 0.08 | 0.13 | 0.24** | 0.17 | 0.33 | 0.23** | 0.161 | 0.321 |
| **Individual- level factors** | | | | | | | | | | | | |
| Age | | | | | | | 0.99** | 0.99 | 0.99 | 0.99** | 0.987 | 0.993 |
| Gender[a] | | | | | | | 0.97 | 0.92 | 1.02 | 1.04 | 0.985 | 1.088 |
| Separated or divorced[b] | | | | | | | 2.35** | 2.19 | 2.53 | 2.35** | 2.188 | 2.529 |
| Widowed[b] | | | | | | | 2.32** | 2.17 | 2.48 | 2.32** | 2.174 | 2.476 |
| Single or never married[b] | | | | | | | 2.80** | 2.54 | 3.07 | 2.80** | 2.544 | 3.072 |
| High Education[c] | | | | | | | 0.96 | 0.90 | 1.02 | 0.96 | 0.897 | 1.017 |
| Working[d] | | | | | | | 0.70** | 0.65 | 0.75 | 0.70 | 0.649 | 0.748 |
| Low FL[e, h] | | | | | | | 1.45** | 1.34 | 1.57 | 1.45 | 1.339 | 1.566 |
| Moderate FL[e, h] | | | | | | | 1.64** | 1.47 | 1.82 | 1.64 | 1.474 | 1.822 |
| Hight FL[e, h] | | | | | | | 2.24** | 1.90 | 2.64 | 2.24 | 1.903 | 2.64 |
| Depressive Mood[f] | | | | | | | 3.40** | 3.22 | 3.58 | 3.40 | 3.221 | 3.583 |
| Pain[g] | | | | | | | 1.18** | 1.12 | 1.24 | 1.18 | 1.118 | 1.239 |
| SPH[i] | | | | | | | 0.71** | 0.69 | 0.73 | 0.71 | 0.689 | 0.728 |
| Country-level factors | | | | | | | | | | | | |
| GINI | | | | 1.39** | 1.10 | 1.75 | 1.52** | 1.17 | 1.97 | 1.52 | 1.17 | 1.97 |
| GDP | | | | | | | | | | 1.04 | 0.86 | 1.26 |
| **Random Effects** | | | | | | | | | | | | |
| var(age) | | | | 0.001 | 0.001 | 0.003 | 0.000 | 0.000 | 0.001 | 0.000 | 0.000 | 0.001 |
| var(cons) | 0.283 | 0.144 | 0.559 | 0.222 | 0.11 | 0.447 | 0.284 | 0.142 | 0.566 | 0.282 | 0.141 | 0.561 |
| ICC | 0.079 | | | | | | | | | | | |
| M&Z r2 | | | | 0.029 | | | 0.243 | | | 0.242 | | |
| AIC | 56897.83 | | | 56596.83 | | | 49114.3 | | | 49116.16 | | |
| Chi2 | 2275.82 | | | 1730.43 | | | 2011.93 | | | 2003.58 | | |
| p-value | <0.001 | | | <0.001 | | | <0.001 | | | <0.001 | | |

**Notes**. Ref categories. [a]Men. [b]Married or partnered.

[c]College and above.

[d]No Worker.

[e]No limitation.

[f]No depressive mood.

[g]Pain

[h]FL stands for Functional limitation.

[i]Self perceived health.

* p<0.05

** p<0.01

***p<0.001. Countries observations were from 1,512 to 7,932 (mean = 4,495.5)

Older adults living in more economically unequal countries were more likely to report loneliness ($OR_{Model2}$ = 1.39; 95% CI: 1.10–1.75). The relationship between country-level economic inequality and loneliness was independent of individual-level compositional factors and country-standardised GDP (Model 4). GDP did not have a statistically significant relationship with loneliness and did not improve the model fit or explained variance; therefore, Model 3 was the best solution for explaining the prevalence of loneliness.

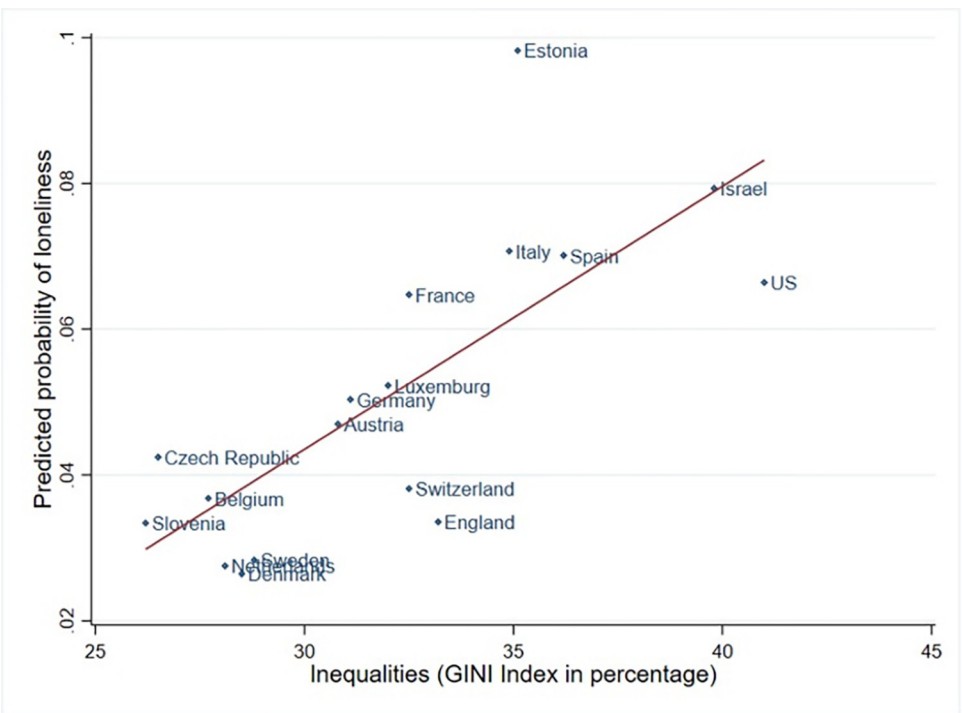

**Fig 2. Country predicted probabilities for loneliness and inequality level.** Note: Probability of loneliness based on model 4.

Model 3 explained 24% of the variation of loneliness (M&Z $R^2$ = 0.243). A unit-increment increase in average economic inequality increased the odds of loneliness by 53% (OR: 1.52; 95% CI: 1.17–1.97). Work status, higher age, and self-reported health decreased the probability of loneliness among the older adults.

Marital status was related to the probability of loneliness. Divorced, widowed or single older adults had 2.35 (95% CI: 2.19–2.53), 2.32 (95% CI: 2.17–2.48) and 2.80 (95% CI: 2.54–3.07) times the odds of experiencing loneliness, respectively, compared to those who had a partner or spouse. Higher functional limitations increased the odds of loneliness. Older adults with low, moderate, or severe functional limitations had 1.45 (95% CI: 1.34–1.57), 1.64 (95% CI: 1.47–1.82), and 2.24 (95% CI: 1.90–2.64) times the odds of experience loneliness, respectively, compared to those with no functional limitations. Depressive mood was a strong predictor of loneliness. People who declared having depressive mood had 3.40 (95% CI: 3.22–3.58) times the odds of loneliness compared to those who did not. Finally, those who reported pain had 1.18 (95% CI: 1.12–1.24) times the odds of loneliness compared to those who did not.

The robustness check confirmed the model results (S3 Table). Fig 2 depicts the positive relationship between the average predicted prevalence of loneliness and country inequality based on model 3.

## 4. Discussion

The present study explored the relationship between country-level income inequality and the prevalence of loneliness in the USA and 16 European countries. Economic inequality within countries was positively associated with loneliness. These results remained consistent after adjusting for sociodemographic characteristics, health status and gross domestic product per capita (GDP).

To our knowledge, this is the first study analysing the relationship between income inequality and loneliness prevalence in the US and 16 European Countries.

## 4.1 Country economic inequality and loneliness

At a country level, the GINI coefficient was positively associated with the prevalence of loneliness. De Jong Gierveld & Tesch-Romer [33] developed an integrative theoretical model that explained loneliness as a result of the combination of individual-level factors and country-level or structural factors [33]. Country-level factor associations with loneliness have previously been explored, primarily comparing individualistic and collectivist societies [59] and Western and Eastern cultural differences in Europe [60]. Reassuringly, in a recently-published analysis of the country-level factors associated with loneliness in eleven countries in Europe, a relationship between economic inequality and loneliness was found [46]. Potential explanations for the relationship between inequality and loneliness included a direct pathway related to socio-economic resources and quality of living conditions and an indirect pathway that considers low social integration, lack of community trust, and a high perception of relative deprivation [60].

The Marmot reports on health inequalities written for the World Health Organization have shown that country-level inequality produces differences in the material conditions within countries and increases relative deprivation, affecting people psychologically [44, 45, 61, 62]. More unequal countries have a steeper social gradient, which means that the social determinants of health have a greater impact [44, 61–65]. Country-level economic inequality directly impacts education, work, income, access to health, and social connections and increases the proportion of people living in poverty [66]. Poor living conditions push more vulnerable people into greater risk of loneliness because of their limited integration into social activities and a lack of social and community support [62]. The Plan of Action for a Decade of Healthy Ageing [41] and the sustainable development goals [42] have set reducing economic inequality within and among countries as one of their priorities. Among other actions, they call for the involvement of all sectors in reducing inequality and for countries to approve social protection policies and improve their regulations of the global financial market and institutions. Previously, it has been highlighted that regardless of a country's economic system, policies and plans should be in place to protect those at bottom of the economic gradient [45].

Individual-level interventions have shown effectiveness in addressing loneliness [67]. However, based on the multilevel composition of loneliness, structural interventions seem to be necessary. National programs targeting people at greater risk of social isolation and loneliness might help overcome inequalities in the distribution of loneliness. Several countries have already implemented programmes addressing social isolation and loneliness in older adults. For instance, European countries have used primary care and other organizations to connect older adults with one another (e.g. Befriending Networks in Ireland, MONALISA in France, the Campaign to End Loneliness in the UK [68, 69]. The United Kingdom has declared social isolation and loneliness as a serious public health problem and has established structural approaches to address them. A series of measures to tackle social isolation and loneliness have been implemented in the last decade, including the creation of a "social prescription" program recently launched by the new Ministry of Loneliness that consist in personalized plans and trains workers to link people with social integration. In the case of the US, although there is no clear national strategy and more efforts might be found through state-based approaches, there are important initiatives like the National Resource Center for Engaging Older Adults [69].

Finally, while the relationship between income inequality and loneliness remained after taking GDP into account, the results of this study challenge the relationship between GDP and

loneliness. A possible explanation can be the low variability in the gross domestic product in the analytic sample. Both, Layte [63] and Tapia-Granados [66] concluded that among high-income countries, it is income inequality and relative poverty, rather than GDP, which impact health outcomes.

## 4.2 Individual factors and loneliness

At the individual level, independent of country-level factors, marital status has a strong posi-tive association with loneliness. People in partnerships have previously reported lower levels of loneliness [15, 21, 23, 70, 71]. Partnerships are strongly related to emotional attachment and social interaction, reducing the levels of emotional and social loneliness [72]. However, changes in marital status and relationships satisfaction also need to be taken into account [71]. Several studies have reported that unsatisfactory or poor-quality relationships are associated with higher loneliness levels among people in partnerships [3, 73, 74].

The current study results also showed that older adults who do not have paid employment were at a higher risk of experiencing loneliness. There is a need for further exploration of the relationship between work status and loneliness. However, most of the older adults in this study were retired or not seeking work. Previous studies have found that retirement neither increases loneliness nor affects health status if older adults have good social connections and support, and plan post-retirement activities [75, 76]. The relationship between work status and loneliness can be linked to a scarcity of economic resources, a reduction in social contacts, and a lack of purpose in life [77].

Health status and self-reported health were strongly related to loneliness. Functional limita-tions, depressive mood, and the presence of pain have been previously reported as factors asso-ciated with increased loneliness among older adults [4, 6, 34, 37, 70, 71, 78]. Accordingly, special attention should be paid to the emotional and social support of those living with severe functional limitations, feeling depressed, or experiencing pain.

Depression has sometimes been studied as a risk factor for loneliness [21]. At the same time, depressed people often feel lonelier [79]. We used depressive mood as a proxy for depres-sion in order to separate depressive symptoms from loneliness experience and avoid multicol-linearity. In line with previous evidence, we found an independent relationship between depressive mood and loneliness prevalence. Finally, self-reported health has been previously related to several health outcomes, including loneliness [76, 80], and the present study showed that good self-reported health is associated with a lower prevalence of loneliness.

Contrary to much previous evidence, gender was not significantly associated with loneli-ness prevalence in this analysis. Previous studies have not accounted for country-level factors. Therefore, the relationship between gender and loneliness may be an expression of older adults' living conditions.

## 4.3 Limitations and future research

The results of the present study should be interpreted in the light of some limitations. First, the cross-sectional associations do not imply causality. Second, unmeasured individual- and coun-try-level factors may bias our results. Though me measured marital status and work status, future research should consider a specific measure for social isolation and non-pension wealth. Third, although missing data in our study was low (<10%), they were not missing completely at random, which may result in selection bias. We performed a bootstrap analysis to address potential bias of our point estimates due to missing information and the precision of our stan-dard errors given the number of clusters in the study. Fourth, measurement bias could be pres-ent given the use of self-reported questionnaires. Even using an indirect measure of loneliness,

the stigma of declaring oneself as "lonely" could have biased participant responses. According to de Jong Gierveld [81], this type of stigma affects men more than women. Future studies should include longitudinal data and different geographical units, and should consider adjusting the estimates for psychological variables (e.g., personality traits, self-esteem, and coping mechanisms), social variables (e.g., social isolation, quality of social connections and relationships and the number of people living at home), and economic variables (e.g., income and wealth measured at the individual and household-level, as well as relative poverty). Harmonizing these variables across countries, however, is not a trivial task and can lead to substantial amounts of missing data and numerous comparability issues.

The current results are important because they provide the impetus to explore the role of country-level income inequality in the prevalence of loneliness further. Addressing the existing gap in wealth distribution may provide an opportunity to improve older adults' wellbeing and life expectancy by reducing loneliness prevalence.

## Supporting information

**S1 File. Predictive models for the prevalence of loneliness.**
(DOCX)

**S1 Table. Analytic sample description.**
(DOCX)

**S2 Table. Missing data.**
(DOCX)

**S3 Table. Observed Bootstrap Normal for model D (N = 75,891).**
(DOCX)

**S4 Table. Reliability analysis of the three items from the R-UCLA scale.**
(DOCX)

## Author Contributions

**Conceptualization:** Thamara Tapia-Muñoz, Ursula M. Staudinger, Esteban Calvo.

**Data curation:** José T. Medina, Esteban Calvo.

**Formal analysis:** Thamara Tapia-Muñoz, Kasim Allel.

**Funding acquisition:** Thamara Tapia-Muñoz, Claudia Miranda-Castillo, Esteban Calvo.

**Writing – original draft:** Thamara Tapia-Muñoz.

**Writing – review & editing:** Thamara Tapia-Muñoz, Ursula M. Staudinger, Kasim Allel, Andrew Steptoe, Claudia Miranda-Castillo, José T. Medina, Esteban Calvo.

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
