## [Decision Letter · Decision Letter 0]

23 May 2022

PONE-D-22-10438Income inequality and its relationship with loneliness prevalence: A cross-sectional study among older adults in the US and 16 European countriesPLOS ONE

Dear Dr. Calvo,

Thank you for submitting your manuscript to PLOS ONE. After careful consideration, we feel that it has merit but does not fully meet PLOS ONE’s publication criteria as it currently stands. Therefore, we invite you to submit a revised version of the manuscript that addresses the points raised during the review process. Please address the comments provided by the reviewers and myself. 

We look forward to receiving your revised manuscript.

Kind regards,

Zhuo Chen, Ph.D.

Academic Editor

PLOS ONE

Journal Requirements:

Additional Editor Comments:

This paper addressed an important topic. However, the reviewers did point out limitations that are critical and should be addressed. I agree with reviewer 2 that the variation of income inequality within a bit more than a dozen countries could be limited. The authors also did not provide a clear description of the causal pathways of the income inequality to loneliness. The level of geographic unit could be relevant as well -- and is relevant to the policy implications. A panel (or longitudinal) analysis may provide more support -- culture and geography could be relevant in the reporting of loneliness.

Reviewers' comments:

Reviewer's Responses to Questions

**Comments to the Author**

1. Is the manuscript technically sound, and do the data support the conclusions?

Reviewer #1: Yes

Reviewer #2: No

2. Has the statistical analysis been performed appropriately and rigorously? 

Reviewer #1: Yes

Reviewer #2: I Don't Know

3. Have the authors made all data underlying the findings in their manuscript fully available?

Reviewer #1: Yes

Reviewer #2: Yes

4. Is the manuscript presented in an intelligible fashion and written in standard English?

Reviewer #1: Yes

Reviewer #2: No

5. Review Comments to the Author

Reviewer #1: Thank you for writing this interesting piece. overall this is good research but there are some areas of clarification that are needed.

PLease review the abstract as it seems that there are key findings from manuscript that should be in manuscript. These comments i make by line # below. Mainly the conclusion takks about addressing income distribution which is a population level solution and not an individual solutions so this is a big "ask" and not quite actionable for decades. So this makes it harder to know what to do with this data.

Detailed comments and questions:

line 35: over what time period the 4.2% to 34%?

36: this is an interesting part of the intro. It seems that you are assuming that social deprivation is a predictor of loneliness without showing the evidence for that. The relationship to isolation is easier to understand than that for loneliness.

50: Person characteristics--wonder if better to say individual sociodemographics?

58: is it really true there is no consesus? The 2 references are from the same author. so it seems more like one authors view rather than there being no consensus. and is it that there is no consensus or is it because of our differences in measurement and definitions?

64: same comment as above. are the cross-country differences also measurement issues?

67: please explain why and how those 3 groups were chosen?

74: non-europeans may need an explanation as to what countries are in northern europe.

87: "little is now about country-level aggreggate factors and loneliness" this is what is missing from abstract

94: this point "welfare stayes..." is better stated than in abstract. consider revising abstract.

96: please state what "marmot studies are?

103-104: this sentence is not clear. consider revising.

105: life expectancy decreased in all groups or by income race/ethnicity. As you likely know in the US there is a strong correlation between income and ethnicity...

108-9: This point still needs needs more clarification as it seems more related to isolation and less to loneliness. OR does it contribute more to one aspect of loneliness. i.e. the structural factors of loneliness (otherwise I dont see how this income inequality contributes to the emotional or functional aspects of loneliness).

167: please explain why recording all to over 90. There is quite a bit of heterogeneity w aging and the oldest old are an important category so please justify this.

167: marital status: are there questions on relationships status (or just marriage?). As this is limiting and at times preferences heterosexuals or those that can legally marry.

170: how were functional measures defined or measured?

216: the individual factors contributing to 91% of total variation is important and also something missing from abstract, or what makes the abstract hard to grasp.

230: dont think its clear how or why the different models were developed. please explain or maybe a summary table?

234: not clear if by work status you mean "answering yes, to working"

268: might highlight not just mental health, but loneliness specifically.

270: typo. "inti" please also provide a theoretical frameowkr as more activity doesnt always equate to less loneliness.

279: nice example of the national resource center. But not sure this is widely used in US. Instead, I might suggest that though there are resources in the US there isnt actually a national strategy.

281-283: sorry for redundancy, but it still feels like you need to go more indepth for a theoretical framework, as the link to isolation is easier to make than loneliness. and how could one actually tackle the dispersion of income distribution? In the US, it would involve dismantling capitalism, and core american individualistic principles. I realize this is an extreme view, and thus, there needs to be a more nuances discussion of what the findings of this paper actually suggest.

300: I wonder if the work status and loneliness is centered on "life purpose" as a mediator.

311: not sure it is correct to define loneliness as a symptom of depression. ie. it isnt part of our standard screenings questions (i.e. phq-9). may be more accurate to stay that loneliness may be an experience that people w depression have. and remember that most lonely people are not depressed.

336-339> im left wondering what now, and HOW do we address the gap in wealth distribution? The conclusion needs to be strengthened.

Reviewer #2: This is a referee report on the paper “Income inequality and its relationship with loneliness prevalence: A cross-sectional study among older adults in the US and 16 European countries”. The authors tried to show a significant association of the country-level index of inequality represented by GINI on individual-level loneliness by using a multilevel logistic regression model. However, it is uncertain whether those analyses support their conclusion.

Please, kindly find the attached file to improve the manuscript.

6. PLOS authors have the option to publish the peer review history of their article (what does this mean?). If published, this will include your full peer review and any attached files.

Reviewer #1: No

Reviewer #2: **Yes: **Yugo Shobugawa

---

## [Author Response · Author response to Decision Letter 0]

29 Jul 2022

LETTER OF RESPONSE ID PONE-D-22-10438:

“Income inequality and its relationship with loneliness prevalence: A cross-sectional study among older adults in the US and 16 European countries”

Dear Zhuo Chen, Ph.D.

Academic Editor, Plos One

We would like to thank you and the reviewers for your thorough and constructive comments on our paper and for the opportunity to revise and resubmit to Plos One. We have considered and worked through these comments with great attention and dedication, bearing in mind the word limit for our article category and the use of new articles that have recently been published in the literature. We uploaded the revised manuscript and supplementary material with and without track changes. We believe our paper is now considerably stronger based on this work. We outline a point-by-point explanation to the provided comments below.

JOURNAL REQUIREMENTS AND EDITOR COMMENTS

 RESPONSE: We carefully followed the style requirements using the style templates.

2. Please provide additional details regarding participant consent. In the ethics statement in the Methods and online submission information, please ensure that you have specified (1) whether consent was informed and (2) what type you obtained (for instance, written or verbal, and if verbal, how it was documented and witnessed). If your study included minors, state whether you obtained consent from parents or guardians. If the need for consent was waived by the ethics committee, please include this information. If you are reporting a retrospective study of medical records or archived samples, please ensure that you have discussed whether all data were fully anonymized before you accessed them and/or whether the IRB or ethics committee waived the requirement for informed consent. If patients provided informed written consent to have data from their medical records used in research, please include this information.

 RESPONSE: Done. See details in the methods section lines 156 to 160. 

 RESPONSE: At the moment of publication, the fully anonymized minimal dataset will be available at the following link: https://github.com/ThammyTapia/loneliness.crosscountry. 

Additional Editor Comments: This paper addressed an important topic. However, the reviewers did point out limitations that are critical and should be addressed. I agree with reviewer 2 that the variation of income inequality within a bit more than a dozen countries could be limited. The authors also did not provide a clear description of the causal pathways of the income inequality to loneliness. The level of geographic unit could be relevant as well -- and is relevant to the policy implications. A panel (or longitudinal) analysis may provide more support -- culture and geography could be relevant in the reporting of loneliness.

 RESPONSE: Secondary analyses of available datasets often name the number of countries (clusters) available as a limitation. We now explicitly acknowledge and address this limitation (see lines 388 to 390 in the limitations section and lines 218 to 220 in the statistical analysis section). Following evidence that the number of clusters and sample sizes for multilevel analyses affected only the standard errors but not the point estimates, we added a bootstrap analysis. To address this limitation, we repeated our final model (model 3) using a hierarchical logistic regression using bootstrap errors with 100 iterations. The results were highly consistent after obtaining more precise standard errors for the first and second levels of analysis. We added this explanation to the methods section lines 271 and the results to the supplementary material section 4. As an additional sensitivity analysis, we conducted logistic regressions ignoring the cluster structure of the data but including countries as a dummy variable in the model. As seen in the output below, the model: (1) overestimated the relationship between country inequality and the prevalence of loneliness, and (2) dropped a country because of the collinearity. In preparing this response we considered the following references: 

• Bryan, M. L., & Jenkins, S. P. (2013). Regression analysis of country effects using multilevel data: A cautionary tale.

• Peter C. Austin & George Leckie (2018) The effect of number of clusters and cluster size on statistical power and Type I error rates when testing random effects variance components in multilevel linear and logistic regression models, Journal of Statistical Computation and Simulation, 88:16, 31513163, 

DOI: 10.1080/00949655.2018.1504945

We also substantially revised the introduction and discussion sections to provide a clearer description of the causal pathways linking income inequality and loneliness, bearing in mind the conceptual differences between social isolation and mental health.

Finally, we highlighted that the availability of longitudinal data at different levels of geographic unit may be a relevant aspect to explore in future research.

 RESPONSE: Done.

COMMENTS BY REVIEWER 1

1. Is the manuscript technically sound, and do the data support the conclusions?

The manuscript must describe a technically sound piece of scientific research with data that supports the conclusions.

Experiments must have been conducted rigorously, with appropriate controls, replication, and sample sizes. The conclusions must be drawn appropriately based on the data presented.

Reviewer #1: Yes

2. Has the statistical analysis been performed appropriately and rigorously?

Reviewer #1: Yes

3. Have the authors made all data underlying the findings in their manuscript fully available?

Reviewer #1: Yes

4. Is the manuscript presented in an intelligible fashion and written in standard English?

Reviewer #1: Yes

 RESPONSE: We thank the reviewer for these positive assessments.

5. Review Comments to the Author

Thank you for writing this interesting piece. overall this is good research but there are some areas of clarification that are needed.

PLease review the abstract as it seems that there are key findings from manuscript that should be in manuscript. These comments i make by line # below. Mainly the conclusion takks about addressing income distribution which is a population level solution and not an individual solutions so this is a big "ask" and not quite actionable for decades. So this makes it harder to know what to do with this data.

 RESPONSE: We thank the reviewer for pointing out several avenues to improve our paper. We worked through these and other comments throughout the manuscript and provided more detailed answers to each comment below.

line 35: over what time period the 4.2% to 34%?

 RESPONSE: We appreciate the reviewer’s comment and clarified this information by including updated numbers from a metanalysis that reviewed pooled data between 2000 and 2019. See abstract lines 34 and introduction section lines 81. In discussing these findings, we considered the following evidence:

• Surkalim, D. L., Luo, M., Eres, R., Gebel, K., van Buskirk, J., Bauman, A., & Ding, D. (2022). The prevalence of loneliness across 113 countries: systematic review and meta-analysis. BMJ (Clinical research ed.), 376, e067068. https://doi.org/10.1136/bmj-2021-067068

36: this is an interesting part of the intro. It seems that you are assuming that social deprivation is a predictor of loneliness without showing the evidence for that. The relationship to isolation is easier to understand than that for loneliness.

 RESPONSE: We appreciate this insightful comment and have rewritten part of the introduction section accordingly. The integrative theoretical model of loneliness developed by Fokkema, T., De Jong Gierveld, J., & Dykstra, P. A. (2012) postulates that loneliness is a multicomponent and multilevel phenomenon resulting from the interaction between individual- and macro-level factors. Reassuringly, interventions at the individual-level failed to explain the cross-country differences in loneliness, and interventions that targeted only individual-level aspects have not been entirely effective.

Health inequalities have been described previously for several health outcomes including loneliness. Health inequality seems to have a direct and indirect relationship with loneliness. The indirect path works by reducing social integration due to a decrease in community trust and an increase in relative deprivation. Social deprivation has been previously linked to loneliness, assuming a health inequality framework. Victor (2020), using data from ELSA and the English Deprivation Index, has reported evidence of the relationship between social deprivation and loneliness in older adults in the UK.

Importantly, social isolation is related to loneliness. While social isolation is the objective measure for a lack of social connectedness or interaction, loneliness is a subjective experience. Social isolation is a risk for loneliness. Even though not all people who experience loneliness are socially isolated, socially isolated people experience higher levels of loneliness. Therefore, initiatives targeting social isolation also have an impact on loneliness.

We considered the following evidence:

• Aartsen, M., Morgan, D., Dahlberg, L., Waldegrave, C., Mikulionienė, S., Rapolienė, G., & Lamura, G. (2020). Exclusion From Social Relations and Loneliness: Individual and Country-Level Changes. Innovation in Aging, 4(Suppl 1), 712–713. https://doi.org/10.1093/geroni/igaa057.2509

• Allen, J., Balfour, R., Bell, R., & Marmot, M. (2014). Social determinants of mental health. International review of psychiatry, 26(4), 392-407.

• de Jong Gierveld, J., & Tesch-Römer, C. (2012). Loneliness in old age in Eastern and Western European societies: theoretical perspectives. European journal of ageing, 9(4), 285–295. https://doi.org/10.1007/s10433-012-0248-2

• de Jong Gierveld, J., Tilburg, T., & Dykstra, P. (2018). New Ways of Theorizing and Conducting Research in the Field of Loneliness and Social Isolation. In A. Vangelisti & D. Perlman (Eds.), The Cambridge Handbook of Personal Relationships (Cambridge Handbooks in Psychology, pp. 391-404). Cambridge: Cambridge University Press. Doi:10.1017/9781316417867.031

• Desa U: Transforming our world: The 2030 agenda for sustainable development. 2016.

• Dykstra P. A. (2009). Older adult loneliness: myths and realities. European journal of ageing, 6(2), 91–100. https://doi.org/10.1007/s10433-009-0110-3

• Fokkema, T., De Jong Gierveld, J., & Dykstra, P. A. (2012). Cross-national differences in older adult loneliness. The Journal of psychology, 146(1-2), 201-228.

• Morgan, D. et al. (2021). Revisiting Loneliness: Individual and Country-Level Changes. In: Walsh, K., Scharf, T., Van Regenmortel, S., Wanka, A. (eds) Social Exclusion in Later Life. International Perspectives on Aging, vol 28. Springer, Cham. https://doi.org/10.1007/978-3-030-51406-8_8

• Marmot, M. (2020). Society and the slow burn of inequality. The lancet, 395(10234), 1413-1414.

• Marmot, M. (2020). Health equity in England: the Marmot review 10 years on. Bmj, 368.

• World Health Organization. (2020). Decade of healthy ageing: baseline report.

• World Health Organization. (2020). Decade of healthy ageing: Plan of action. Proceedings of the 73rd World Health Assembly, Geneva, Switzerland, 17-21.

• Victor, C.R. and Pikhartova, J. (2020) ‘Lonely places or lonely people? Investigating the relationship between loneliness and place of residence’, BMC Public Health, 20, 778, pp. 1-12. Doi: 10.1186/s12889-020-08703-8.

50: Person characteristics—wonder if better to say individual sociodemographics?

 RESPONSE: We replaced “person characteristics” with “individual sociodemographics”. 

58: is it really true there is no consensus? The 2 references are from the same author. So it seems more like one authors view rather than there being no consensus. And is it that there is no consensus or is it because of our differences in measurement and definitions?. 

 RESPONSE: We agree with this comment, deleted the controversial text, and added a brief review on definitions of loneliness in lines 58 to 60.

64: same comment as above. Are the cross-country differences also measurement issues?. 

 RESPONSE: The three surveys (ELSA, HRS, SHARE) used in our study are nationally representative surveys aimed at achieving cross-country comparability. They measured loneliness using the same 3-item R-UCLA scale. Other studies comparing countries within Europe also found cross-country differences, with Northern Europe presenting lower levels of loneliness. Importantly, cross-country differences remain evident after adjusting the models by type of measurement. A recent metanalysis showed cross-country differences in loneliness adjusting the type of measurement. To support these arguments, we reviewed the following evidence: 

• Aartsen, M., Morgan, D., Dahlberg, L., Waldegrave, C., Mikulionienė, S., Rapolienė, G., & Lamura, G. (2020). Exclusion From Social Relations and Loneliness: Individual and Country-Level Changes. Innovation in Aging, 4(Suppl 1), 712–713. https://doi.org/10.1093/geroni/igaa057.2509

• Buecker, S., Maes, M., Denissen, J. J. A., & Luhmann, M. (2020). Loneliness and the Big Five Personality Traits: A Meta–Analysis. European Journal of Personality, 34(1), 8–28. https://doi.org/10.1002/per.2229

• Dykstra PA. Older adult loneliness: myths and realities. Eur J Ageing. 2009 Jun;6(2):91-100. doi: 10.1007/s10433-009-0110-3. Epub 2009 Apr 4. PMID: 19517025; PMCID: PMC2693783.

• Fokkema, T., De Jong Gierveld, J., & Dykstra, P. A. (2012). Cross-national differences in older adult loneliness. The Journal of psychology, 146(1-2), 201-228.

• Surkalim, D. L., Luo, M., Eres, R., Gebel, K., van Buskirk, J., Bauman, A., & Ding, D. (2022). The prevalence of loneliness across 113 countries: systematic review and meta-analysis. BMJ (Clinical research ed.), 376, e067068. https://doi.org/10.1136/bmj-2021-067068

• Yang, K., & Victor, C. (2011). Age and loneliness in 25 European nations. Ageing & Society, 31(8), 1368–1388. https://doi.org/10.1017/S0144686X1000139X

67: please explain why and how those 3 groups were chosen?

 RESPONSE: The authors cited claim that “The grouping was a result of studying the relationship for each individual nation (details not shown here due to limited space).” Consequently, we now say in line 73 that “Countries were divided into three groups based on the author’s assessment of the relationship for each country”. In the first manuscript submitted, we re-arranged the number assigned to the groups from lowest to the highest prevalence, but to avoid confusion we now present groups using their original number (lines 72 to 79). We considered the following reference: 

• Surkalim, D. L., Luo, M., Eres, R., Gebel, K., van Buskirk, J., Bauman, A., & Ding, D. (2022). The prevalence of loneliness across 113 countries: systematic review and meta-analysis. BMJ (Clinical research ed.), 376, e067068. https://doi.org/10.1136/bmj-2021-067068

74: non-europeans may need an explanation as to what countries are in northern europe.

 RESPONSE: We added the list of countries within the regions mentioned in lines 74 to 81. We considered the following reference:

• Surkalim, D. L., Luo, M., Eres, R., Gebel, K., van Buskirk, J., Bauman, A., & Ding, D. (2022). The prevalence of loneliness across 113 countries: systematic review and meta-analysis. BMJ (Clinical research ed.), 376, e067068. https://doi.org/10.1136/bmj-2021-067068

87: "little is now about country-level aggreggate factors and loneliness" this is what is missing from abstract.

 RESPONSE: This is an excellent point. We have added the information to the abstract in line 37.

94: this point "welfare stayes..." is better stated than in abstract. consider revising abstract.

 RESPONSE: Following this and the previous comment, we revised the abstract, which now says that “The relationship between country-level factors and loneliness, however, has been underexplored.”

96: please state what "marmot studies are?

 RESPONSE: The Marmot reports are classic studies on health inequalities conducted by Sir Michael Marmot for the government of the UK or for the World Health Organization. We mentioned them as “Marmot reports” and referenced the specific reports wherever appropriate. For clarification we have now added the subject of the studies “health inequalities” where we mentioned them. Please see a summary of the Marmot studies below: 

Michael Marmot has led research groups on health inequalities. He was chair of the following commissions:

• The Commission on Social Determinants of Health (CSDH), at the World Health Organization. The results were synthetized in the Report: Marmot, M., Friel, S., Bell, R., Houweling, T. A., Taylor, S., & Commission on Social Determinants of Health. (2008). Closing the gap in a generation: health equity through action on the social determinants of health. The lancet, 372(9650), 1661-1669.

• The Regional Commission on the Social Determinants of Health at WHO to review health inequities in WHO’s Eastern Mediterranean Region. The results and conclusions can be found in the document: Marmot, M., Al-Mandhari, A., Ghaffar, A., El-Adawy, M., Hajjeh, R., Khan, W., & Allen, J. (2021). Build back fairer: achieving health equity in the Eastern Mediterranean region of WHO. The Lancet, 397(10284), 1527-1528.

• The Commission on Equity and Health Inequalities in the Americas at the Pan-American Health Organization dependent of the World Health Organization (PAHO/ WHO). Final report: Commission of the Pan American Health Organization on Equity and Health Inequalities in the Americas. (2019). Just Societies: Health Equity and Dignified Lives. Report of the Commission of the Pan American Health Organization on Equity and Health Inequalities in the Americas.

The British Government and the World Health Organization also requested him to conduct reviews of Health Inequalities in England, originating the following reports: 

• Marmot, M. (2013). Fair society, healthy lives. Fair society, healthy lives, 1-74.

• Marmot, M., Allen, J., Bell, R., Bloomer, E., & Goldblatt, P. (2012). WHO European review of social determinants of health and the health divide. The Lancet, 380(9846), 1011-1029.

• Marmot, M. (2020). Health equity in England: the Marmot review 10 years on. Bmj, 368 (an update of the Fair society, healthy lives report)

• Al-Mandhari, A., Marmot, M., Abdu, G., Hajjeh, R., Allen, J., Khan, W., & El-Adawy, M. (2021). COVID-19 pandemic: a unique opportunity to ‘build back fairer’and reduce health inequities in the Eastern Mediterranean Region. Eastern Mediterranean Health Journal, 27(3), 217-219.

As head of the UCL Department of Epidemiology & Public Health Marmot led the Whitehall II Studies of British Civil Servants and the English Longitudinal Study of Ageing (ELSA) and published the following books: 

• Marmot, M. (2015). The health gap: The challenge of an unequal world. London: Bloomsbury.

• Marmot, M. (2005). Status syndrome: How your social standing directly affects your health. A&C Black.

103-104: this sentence is not clear. consider revising.

 RESPONSE: Done. Following this and other comments we substantially reviewed the introduction and methods sections. 

105: life expectancy decreased in all groups or by income race/ethnicity. As you likely know in the US there is a strong correlation between income and ethnicity...

 RESPONSE: We appreciate the author’s comment. Although the life expectancy decreased in all groups with different levels depending on ethnicity, we revised the introduction focusing exclusively on evidence related to loneliness. We have considered: 

• Organization WH: Social determinants of mental health. 2014.

• Marmot M: Health equity in England: the Marmot review 10 years on. Bmj 2020, 368.

108-9: This point still needs needs more clarification as it seems more related to isolation and less to loneliness. OR does it contribute more to one aspect of loneliness. i.e. the structural factors of loneliness (otherwise I dont see how this income inequality contributes to the emotional or functional aspects of loneliness).

 RESPONSE: We now offer a much more in-depth description in the introduction section that explicitly addresses the differences between loneliness and social isolation. Loneliness varies across countries, partly due to country-level factors that shape people’s living conditions, access to education, work, and health, possibility for social connexions, household economic level, retirement plans, expectations of social connections, and expectations for retirement years, among other factors related to loneliness. The Integrative Theoretical Model of loneliness postulates that the subjective experience of loneliness results from subjective and objective factors, and from the interaction between micro-level or individual variables with macro-level or social/environmental variables. Older adults living in highly unequal countries are thus expected to experience more loneliness because they will be more socially isolated and have limited access to support. They would be living with multiple chronic diseases and disabilities, in poverty, with poor access to health care, lack of social and leisure opportunities, living in places with higher crime levels, and lower expectations for community-based support. It would also affect trust, self-esteem, and intergenerational social interaction expectations. Loneliness is a common experience and is not detrimental to health in all cases. The evolutionary theory of loneliness establishes loneliness as an adaptative mechanism for species and individual survival. However, when loneliness is chronic (very intense and long-lasting) it manifests as a risk factor for adverse health outcomes. Therefore, analysing the likelihood of people reporting severe loneliness using country-level factors fills knowledge gaps about cross-country differences in loneliness or individual differences in loneliness that cannot be fully attributed to individual-level factors. In updating our manuscript, we have considered the following references: 

• Aartsen, M., Morgan, D., Dahlberg, L., Waldegrave, C., Mikulionienė, S., Rapolienė, G., & Lamura, G. (2020). Exclusion From Social Relations and Loneliness: Individual and Country-Level Changes. Innovation in Aging, 4(Suppl 1), 712–713. https://doi.org/10.1093/geroni/igaa057.2509

• Buecker, S., Maes, M., Denissen, J. J. A., & Luhmann, M. (2020). Loneliness and the Big Five Personality Traits: A Meta–Analysis. European Journal of Personality, 34(1), 8–28. https://doi.org/10.1002/per.2229

• de Jong Gierveld, J., & Tesch-Römer, C. (2012). Loneliness in old age in Eastern and Western European societies: theoretical perspectives. European journal of ageing, 9(4), 285–295. https://doi.org/10.1007/s10433-012-0248-2

• de Jong Gierveld, J., Tilburg, T., & Dykstra, P. (2018). New Ways of Theorizing and Conducting Research in the Field of Loneliness and Social Isolation. In A. Vangelisti & D. Perlman (Eds.), The Cambridge Handbook of Personal Relationships (Cambridge Handbooks in Psychology, pp. 391-404). Cambridge: Cambridge University Press. Doi:10.1017/9781316417867.031

• Dykstra P. A. (2009). Older adult loneliness: myths and realities. European journal of ageing, 6(2), 91–100. https://doi.org/10.1007/s10433-009-0110-3

• Fokkema, T., De Jong Gierveld, J., & Dykstra, P. A. (2012). Cross-national differences in older adult loneliness. The Journal of psychology, 146(1-2), 201-228.

• Hawkley, L. C., & Capitanio, J. P. (2015). Perceived social isolation, evolutionary fitness and health outcomes: a lifespan approach. Philosophical transactions of the Royal Society of London. Series B, Biological sciences, 370(1669), 20140114. https://doi.org/10.1098/rstb.2014.0114

• Morgan, D. et al. (2021). Revisiting Loneliness: Individual and Country-Level Changes. In: Walsh, K., Scharf, T., Van Regenmortel, S., Wanka, A. (eds) Social Exclusion in Later Life. International Perspectives on Aging, vol 28. Springer, Cham. https://doi.org/10.1007/978-3-030-51406-8_8

• Marmot M: Health equity in England: the Marmot review 10 years on. Bmj 2020, 368.

• Mund, M., Freuding, M. M., Möbius, K., Horn, N., & Neyer, F. J. (2020). The Stability and Change of Loneliness Across the Life Span: A Meta-Analysis of Longitudinal Studies. Personality and social psychology review : an official journal of the Society for Personality and Social Psychology, Inc, 24(1), 24–52. https://doi.org/10.1177/1088868319850738

• Organization WH: Social determinants of mental health. 2014.

• Organization WH: Decade of healthy ageing: baseline report. 2020.

• Organization WH: Decade of healthy ageing: Plan of action. Proceedings of the 73rd World Health Assembly, Geneva, Switzerland 2020:17-21.

• Yang, K., & Victor, C. (2011). Age and loneliness in 25 European nations. Ageing & Society, 31(8), 1368–1388. https://doi.org/10.1017/S0144686X1000139X

167: please explain why recording all to over 90. There is quite a bit of heterogeneity w aging and the oldest old are an important category so please justify this.

 RESPONSE: We now explain in line 189 that the recodification was made based on the small number of people who are more than 90 years old (n=600 for the 17 countries: 0.79% from the total sample).

167: marital status: are there questions on relationships status (or just marriage?). 

As this is limiting and at times preferences heterosexuals or those that can legally marry.

 RESPONSE: As we now explain in the manuscript line 191, the measurement includes both marriages and partnerships without restriction to legal unions. 

170: how were functional measures defined or measured?

 RESPONSE: Lines 196 to 198 in the revised manuscript now say that “Functional limitations were assessed using the three items (bathe, dress, and eat) defined in the Wallace and Herzog measure Activities of daily living (ADLs).” We have considered:

• Wallace, R. B., & Herzog, A. R. (1995). Overview of the Health Measures in the Health and Retirement Study. The Journal of Human Resources, 30, S84–S107. https://doi.org/10.2307/146279

216: the individual factors contributing to 91% of total variation is important and also something missing from abstract, or what makes the abstract hard to grasp.

 RESPONSE: We appreciate this comment, which helped us reformulate the results section correcting the ICC interpretation of the updated analyses with the final sample. We reported an ICC of 0.079, which means that 7.9% of the individual-level variance was explained by cross-country differences. We also clarified that the final model (model 3 as a whole) explained 24% of the total variance for the prevalence of loneliness. 

230: don’t think it’s clear how or why the different models were developed. please explain or maybe a summary table?

 RESPONSE: We added a more detailed explanation in the methods section lines 211 to 216, which now says: “We computed four sequential models to analyse the relationship between country-level economic inequality and individual-level loneliness. Model 1 included a fixed and random intercept only, allowing for an estimation of Intra-Class Correlation (ICC). Model 2 included the GINI index, allowing for an unadjusted estimation of its relationship with loneliness. Model 3 added individual-level control variables to model 2. Finally, model 4 added GDP per capita as a country-level control variable to model 3.”

234: not clear if by work status you mean "answering yes, to working"

 RESPONSE: We have added more details to the method section lines xx, which now says: “Work status measured paid work (full- or part-time, salaried or self-employed, combined or not with partial retirement) as opposed to not working for pay (complete retirement, disabled, unemployed, or out of the labour force).”

268: might highlight not just mental health, but loneliness specifically.

 RESPONSE: Following this and other comments, we substantially revised the whole discussion section. 

270: typo. "inti" please also provide a theoretical frameowkr as more activity doesnt always equate to less loneliness. 

 RESPONSE: Done. Following this and other comments, we substantially revised the whole discussion section. 

279: nice example of the national resource center. But not sure this is widely used in US. Instead, I might suggest that though there are resources in the US there isnt actually a national strategy.

 RESPONSE: We appreciate the reviewer’s comment and agree that having resources and a national strategy are two different things. The revised text in lines 340 to 342 say: “In the case of the US, although there is no clear national strategy and more efforts might be found through state-based approaches, there are important initiatives like the National Resource Center for Engaging Older Adults [22].” We have considered the following evidence:

• McDaid, D., Qualter, P., Arsenault, L., Barreto, M., Fett, A. K., Hey, N., ... & Victor, C. (2022). Tackling loneliness evidence review.

• Marmot M: Health equity in England: the Marmot review 10 years on. Bmj 2020, 368.

• Mund, M., Freuding, M. M., Möbius, K., Horn, N., & Neyer, F. J. (2020). The Stability and Change of Loneliness Across the Life Span: A Meta-Analysis of Longitudinal Studies. Personality and social psychology review : an official journal of the Society for Personality and Social Psychology, Inc, 24(1), 24–52. https://doi.org/10.1177/1088868319850738

• Organization WH: Social determinants of mental health. 2014.

• Organization WH: Decade of healthy ageing: baseline report. 2020.

• Organization WH: Decade of healthy ageing: Plan of action. Proceedings of the 73rd World Health Assembly, Geneva, Switzerland 2020:17-21.

• Yang, K., & Victor, C. (2011). Age and loneliness in 25 European nations. Ageing & Society, 31(8), 1368–1388. https://doi.org/10.1017/S0144686X1000139X

281-283: sorry for redundancy, but it still feels like you need to go more indepth for a theoretical framework, as the link to isolation is easier to make than loneliness. and how could one actually tackle the dispersion of income distribution? In the US, it would involve dismantling capitalism, and core american individualistic principles. I realize this is an extreme view, and thus, there needs to be a more nuances discussion of what the findings of this paper actually suggest.

 RESPONSE: We appreciate this comment, which helped us to substantially revise the discussion section. The updated discussion was based on the perspective of health inequalities, social determinants of health, and the integrative model of loneliness. Social relationships are essential for health. Social isolation and loneliness are two distinct aspects of social relationships, not always associated. However, even considering that people who feel lonely are not necessarily isolated, social isolation has been described as being a high risk for loneliness. Interventions usually focus on improving social connectedness to reduce both social isolation and loneliness. The current conclusions about the interventions for loneliness are that: 1) they need to target people more at risk, 2) they need to consider the specific elements of the individual experience of loneliness (age, personality, lack of network, or lack of economic resources), and 3) the individual interventions need to be combined with structural interventions to made communities more interconnected and ageing friendly. 

Loneliness has been previously related to economic inequality and social deprivation. The integrative model of loneliness describes the multilevel structure of loneliness, with individual and macro-social factors interacting to produce living conditions that increase and chronify the level of loneliness.

Studies on health inequalities pointed out that income inequality produces material and subjective deprivation affecting people’s health and mental health. The groups at the bottom of the social gradient have a lower life expectancy and live more years with disability and chronic diseases. We are also exploring how this social gradient affects social relationships, increasing social isolation, and loneliness. In this sense, the international calls to reduce inequality among and within countries are not focusing on changing the economic system but on establishing social protection measures to support countries with lower resources and the most disadvantaged groups within countries with high incomes, like the case of the US. 

We have added the perspectives of the United Nations and the World Health Organization about inequality and healthy ageing, and the integrative model of loneliness, to the discussion section. We also recognise the lack of an objective measure of social isolation in the models as a limitation of this study (see discussion section lines 393 to 398). However, based on the Steptoe Index of Social Isolation, we know that marital status can be used as a proxy. Future research should consider social isolation as a potential confounder of the association between between-country inequality and loneliness.

In preparing this response we have considered the following references: 

• Aartsen, M., Morgan, D., Dahlberg, L., Waldegrave, C., Mikulionienė, S., Rapolienė, G., & Lamura, G. (2020). Exclusion From Social Relations and Loneliness: Individual and Country-Level Changes. Innovation in Aging, 4(Suppl 1), 712–713. https://doi.org/10.1093/geroni/igaa057.2509

• McDaid, D., Qualter, P., Arsenault, L., Barreto, M., Fett, A. K., Hey, N., ... & Victor, C. (2022). Tackling loneliness evidence review.

• Buecker, S., Maes, M., Denissen, J. J. A., & Luhmann, M. (2020). Loneliness and the Big Five Personality Traits: A Meta–Analysis. European Journal of Personality, 34(1), 8–28. https://doi.org/10.1002/per.2229

• de Jong Gierveld, J., & Tesch-Römer, C. (2012). Loneliness in old age in Eastern and Western European societies: theoretical perspectives. European journal of ageing, 9(4), 285–295. https://doi.org/10.1007/s10433-012-0248-2

• de Jong Gierveld, J., Tilburg, T., & Dykstra, P. (2018). New Ways of Theorizing and Conducting Research in the Field of Loneliness and Social Isolation. In A. Vangelisti & D. Perlman (Eds.), The Cambridge Handbook of Personal Relationships (Cambridge Handbooks in Psychology, pp. 391-404). Cambridge: Cambridge University Press. Doi:10.1017/9781316417867.031

• Dykstra P. A. (2009). Older adult loneliness: myths and realities. European journal of ageing, 6(2), 91–100. https://doi.org/10.1007/s10433-009-0110-3

• Fokkema, T., De Jong Gierveld, J., & Dykstra, P. A. (2012). Cross-national differences in older adult loneliness. The Journal of psychology, 146(1-2), 201-228.

• Hawkley, L. C., & Capitanio, J. P. (2015). Perceived social isolation, evolutionary fitness and health outcomes: a lifespan approach. Philosophical transactions of the Royal Society of London. Series B, Biological sciences, 370(1669), 20140114. https://doi.org/10.1098/rstb.2014.0114

• Morgan, D. et al. (2021). Revisiting Loneliness: Individual and Country-Level Changes. In: Walsh, K., Scharf, T., Van Regenmortel, S., Wanka, A. (eds) Social Exclusion in Later Life. International Perspectives on Aging, vol 28. Springer, Cham. https://doi.org/10.1007/978-3-030-51406-8_8

• Marmot M: Health equity in England: the Marmot review 10 years on. Bmj 2020, 368.

• Mund, M., Freuding, M. M., Möbius, K., Horn, N., & Neyer, F. J. (2020). The Stability and Change of Loneliness Across the Life Span: A Meta-Analysis of Longitudinal Studies. Personality and social psychology review : an official journal of the Society for Personality and Social Psychology, Inc, 24(1), 24–52. https://doi.org/10.1177/1088868319850738

• National Academies of Sciences, Engineering, and Medicine. (2020). Social isolation and loneliness in older adults: Opportunities for the health care system. National Academies Press.

• Organization WH: Social determinants of mental health. 2014.

• Organization WH: Decade of healthy ageing: baseline report. 2020.

• Organization WH: Decade of healthy ageing: Plan of action. Proceedings of the 73rd World Health Assembly, Geneva, Switzerland 2020:17-21.

• Steptoe A, Shankar A, Demakakos P, Wardle J. Social isolation, loneliness, and all-cause mortality in older men and women. Proceedings of the National Academy of Sciences 2013; 110(15): 5797-801.

• Yang, K., & Victor, C. (2011). Age and loneliness in 25 European nations. Ageing & Society, 31(8), 1368–1388. https://doi.org/10.1017/S0144686X1000139X

300: I wonder if the work status and loneliness is centered on "life purpose" as a mediator.

 RESPONSE: This is an interesting question. Work is a protective factor for health, and it is related to both social role and personal identity. Income aside (because in most countries with high levels of inequality, retirement pensions do not meet older adults' needs), it might be that life purpose is a mediator between work and loneliness. We have added this insightful idea as a suggestion for future research. We considered the following evidence:

• Bowen CE, Noack MG, Staudinger UM: Chapter 17 - Aging in the Work Context. In: Handbook of the Psychology of Aging (Seventh Edition). Edited by Schaie KW, Willis SL. San Diego: Academic Press; 2011: 263-277.

• Hill P.L., Cardador M.T. (2017) Purpose, Meaning, and Work in Later Life. In: Pachana N.A. (eds) Encyclopedia of Geropsychology. Springer, Singapore. https://doi.org/10.1007/978-981-287-082-7_299

311: not sure it is correct to define loneliness as a symptom of depression. ie. it isnt part of our standard screenings questions (i.e. phq-9). may be more accurate to stay that loneliness may be an experience that people w depression have. and remember that most lonely people are not depressed.

 RESPONSE: Following this and other comments we have reformulated the discussion section to avoid confusion. We agree that people who feel lonely are not necessarily depressed. However, the Center for Epidemiological Studies-Depression (CES-D) include the item "During the past week… I felt lonely" as one of several indicators of depressive symptomatology. We considered the followjng evidence:

• Lewinsohn, P.M., Seeley, J.R., Roberts, R.E., & Allen, N.B. (1997). Center for Epidemiological Studies-Depression Scale (CES-D) as a screening instrument for depression among community-residing older adults. Psychology and Aging, 12, 277- 287.

• Radloff, L. S. (1977). The CES-D scale: A self report depression scale for research in the general population. Applied Psychological Measurements, 1, 385-401.

336-339> im left wondering what now, and HOW do we address the gap in wealth distribution? The conclusion needs to be strengthened.

 RESPONSE: We appreciate the reviewer's comment, which helped us to substantially revise our discussion and conclusion. Although there is no silver bullet to address country-level income inequality, we included recommendations based on the Marmot Reports. For example, we added the following text in lines 323 to 325: “Among other actions, they call for the involvement of all sectors in reducing inequality and for countries to approve social protection policies and improve their regulations of the global financial market and institutions. Previously, it has been highlighted that regardless of a country’s economic system, policies and plans should be in place to protect those at bottom of the economic gradient [45].”

Taking a different angle, in lines 328 we argue that: “Individual-level interventions have shown effectiveness in addressing loneliness [69]. However, based on the multilevel composition of loneliness, structural interventions seem to be necessary. National programs targeting people at greater risk of social isolation and loneliness might help overcome inequalities in the distribution of loneliness. Several countries have already implemented programmes addressing social isolation and loneliness in older adults. For instance, European countries have used primary care and other organizations to connect older adults with one another (e.g. Befriending Networks in Ireland, MONALISA in France, the Campaign to End Loneliness in the UK [70, 71]. The United Kingdom has declared social isolation and loneliness as a serious public health problem and has established structural approaches to address them. A series of measures to tackle social isolation and loneliness have been implemented in the last decade, including the creation of a “social prescription” program recently launched by the new Ministry of Loneliness that consist in personalized plans and trains workers to link people with social integration. In the case of the US, although there is no clear national strategy and more efforts might be found through state-based approaches, there are important initiatives like the National Resource Center for Engaging Older Adults [72].”

COMMENTS BY REVIEWER 2

1. Is the manuscript technically sound, and do the data support the conclusions?

The manuscript must describe a technically sound piece of scientific research with data that supports the conclusions.

Experiments must have been conducted rigorously, with appropriate controls, replication, and sample sizes. The conclusions must be drawn appropriately based on the data presented.

Reviewer #2: No.

 RESPONSE: We updated our analyses and made several changes throughout the manuscript to strengthen our methods, results, and conclusions. Wherever appropriate, we softened the language to ensure that all conclusions were based on the data and results presented.

2. Has the statistical analysis been performed appropriately and rigorously?

Reviewer #2: I Don't Know

 RESPONSE: We believe our updated analyses are appropriate and rigorous and hope that reviewer 2 will agree. We followed STROBE reporting guidelines to ensure that all the details on the strengths and weaknesses of our study can be fully assessed by the reader.

3. Have the authors made all data underlying the findings in their manuscript fully available?

Reviewer #2: Yes

 RESPONSE: We thank the reviewer for this positive assessment.

4. Is the manuscript presented in an intelligible fashion and written in standard English?

Reviewer #2: No

 RESPONSE: We checked carefully for typographical and grammatical errors and made several changes throughout the manuscript to improve the narrative and ensure that our language was clear, correct, and unambiguous.

5. Review Comments to the Author

This is a referee report on the paper “Income inequality and its relationship with loneliness prevalence: A cross-sectional study among older adults in the US and 16 European countries”. The authors tried to show a significant association of the country-level index of inequality represented by GINI on individual-level loneliness by using a multilevel logistic regression model. However, it is uncertain whether those analyses support their conclusion. Please, kindly find the attached file to improve the manuscript.

 RESPONSE: We have considered and worked through these comments with great attention and dedication. We believe our updated analyses are appropriate, rigorous and hope that reviewer 2 will agree that they support our conclusions.

Major comments

Is it appropriate to apply multilevel analysis on the data from 17 countries? The reviewer understands the prevalence of outcome significantly varies across the countries however N=17 is too few for multilevel analysis.

 RESPONSE: Secondary analysis of available datasets often has the number of countries (clusters) availability as a limitation. We now explicitly acknowledge and address this limitation (see lines 388 to 390 in the limitations section and lines 218 to 220 in the statistical analysis section). Following evidence that the number of clusters and sample sizes for multilevel analyses affected only the standard errors but not the point estimates, we added a bootstrap analysis. We repeated our final model (model 3) using a hierarchical logistic regression using bootstrap errors with 100 iterations. The results were highly consistent after obtaining more precise standard errors for the first and second levels of analysis. We added this explanation to the methods section lines 218 to 220 and the results to the supplementary material section 4.

As an additional sensitivity analysis, we conducted logistic regressions ignoring the cluster structure of the data but including countries as a dummy variable in the model. As seen in the output below, the model: (1) overestimated the relationship between country inequality and the prevalence of loneliness, and (2) dropped a country because of the collinearity.

In preparing this response we considered the following references: 

• Bryan, M. L., & Jenkins, S. P. (2013). Regression analysis of country effects using multilevel data: A cautionary tale.

• Peter C. Austin & George Leckie (2018) The effect of number of clusters and cluster size on statistical power and Type I error rates when testing random effects variance components in multilevel linear and logistic regression models, Journal of Statistical Computation and Simulation, 88:16, 31513163, 

DOI: 10.1080/00949655.2018.1504945

Logistic regression results using countries as covariate:

In the main result section (3.2 in the text on pages 12-13), the interpretation of the analyses is uncertain.

 RESPONSE: We substantially revised the interpretation of analyses to avoid lack of clarity and ambiguities. 

-In line 217, the authors mentioned “Country differences accounted for seven percent of the total variation of the being lonely.” It is unclear what is the number “7%”. Also, the next sentence “There was statistically significant variability in the odds of loneliness between the countries” is not understandable based on the OR of constant value in the fixed effect. It is not clear what beta01 means (beta01 is not defined in the equation in the method section).

 RESPONSE: We appreciate this comment and revised the text accordingly, which now says: “HLM results are reported in Table 3. The unadjusted relationship between individual-level variables and loneliness prevalence was statistically significant (see supplementary materials, section 3, Table S3-B). As indicated by the Intra-Class Correlation (ICC), the variability between countries accounted for 7.9% of the total variation in the likelihood of an individual being lonely. In an average country, the odds of being lonely, defined as scoring more than 6 points in the three items of R-UCLA, was 0.13. However, there was statistically significant variability in the odds of loneliness between countries (Between country variance= 0.283; 95% IC: 0.144-0.559).”

Please, unify an expression of terms. For instance, the authors uses Self reported health and self-perceived health. This mixed expression is confusing readers. 

 RESPONSE: Done. 

Table B in S2 is not easy to understand.

 RESPONSE: We updated the table for clarity. 

Minor comments

In line 77 on page 5, R-UCLA suddenly appears. Please, explain what this means at the first use in the text even if the authors explained it in detail after this section.

 RESPONSE: We have added the full name of the scale in the introduction section lines 87 and 88, which now say “revised version of The University of California Los Angeles Loneliness Scale (R-UCLA scale)”

In Fig. 1, in the column of ELSA, the number of bottoms is “N=7934”. Why does the number of participants increase after incomplete cases are excluded?

 RESPONSE: We thank the reviewer for pointing out this error, which we thoroughly corrected in the revised manuscript (see Fig. 1). 

In line 177 on page 9, the authors mentioned: “see Table B in S3 Tables”. Please, mention a result in the result section.

 RESPONSE: Done. 

In Table 1, values of frequency and mean are mixed. Unit of age is not shown, and the value of Self-Reported Health is unclear. These make the table difficult to understand.

 RESPONSE: We updated Table 1 as suggested. 

The numbering of the tables is confusing. There are Table A and B in S2 and S3 Tables. The authors may change to for example “S2-A”.

 RESPONSE: Done.

---

## [Editor Report · Decision Letter 1]

30 Aug 2022

Income inequality and its relationship with loneliness prevalence: A cross-sectional study among older adults in the US and 16 European countries

PONE-D-22-10438R1

Dear Dr. Calvo,

We’re pleased to inform you that your manuscript has been judged scientifically suitable for publication and will be formally accepted for publication once it meets all outstanding technical requirements.

Kind regards,

Zhuo Chen, Ph.D.

Academic Editor

PLOS ONE
---

## [Editor Report · Acceptance letter]

14 Sep 2022

PONE-D-22-10438R1 

Income inequality and its relationship with loneliness prevalence: A cross-sectional study among older adults in the US and 16 European countries 

Dear Dr. Calvo:

I'm pleased to inform you that your manuscript has been deemed suitable for publication in PLOS ONE. Congratulations! Your manuscript is now with our production department. 

Kind regards, 

on behalf of

Prof. Zhuo Chen 

Academic Editor

PLOS ONE